# TIRAP/Mal Positively Regulates TLR8-Mediated Signaling via IRF5 in Human Cells

**DOI:** 10.3390/biomedicines10071476

**Published:** 2022-06-22

**Authors:** Kaja Elisabeth Nilsen, Astrid Skjesol, June Frengen Kojen, Terje Espevik, Jørgen Stenvik, Maria Yurchenko

**Affiliations:** 1Centre of Molecular Inflammation Research, Norwegian University of Science and Technology, NO-7491 Trondheim, Norway; kaja.e.nilsen@ntnu.no (K.E.N.); astrid.skjesol@gmail.com (A.S.); june.f.kojen@ntnu.no (J.F.K.); terje.espevik@ntnu.no (T.E.); jorgen.stenvik@ntnu.no (J.S.); 2Department of Clinical and Molecular Medicine, Norwegian University of Science and Technology, NO-7491 Trondheim, Norway; 3Department of Infectious Diseases, Clinic of Medicine, St. Olavs Hospital HF, Trondheim University Hospital, NO-7006 Trondheim, Norway

**Keywords:** TLR8, TIRAP, Akt, human macrophages, IRF5, Akt inhibitors

## Abstract

Toll-like receptor 8 (TLR8) recognizes single-stranded RNA of viral and bacterial origin as well as mediates the secretion of pro-inflammatory cytokines and type I interferons by human monocytes and macrophages. TLR8, as other endosomal TLRs, utilizes the MyD88 adaptor protein for initiation of signaling from endosomes. Here, we addressed the potential role of the Toll-interleukin 1 receptor domain-containing adaptor protein (TIRAP) in the regulation of TLR8 signaling in human primary monocyte-derived macrophages (MDMs). To accomplish this, we performed *TIRAP* gene silencing, followed by the stimulation of cells with synthetic ligands or live bacteria. Cytokine-gene expression and secretion were analyzed by quantitative PCR or Bioplex assays, respectively, while nuclear translocation of transcription factors was addressed by immunofluorescence and imaging, as well as by cell fractionation and immunoblotting. Immunoprecipitation and Akt inhibitors were also used to dissect the signaling mechanisms. Overall, we show that TIRAP is recruited to the TLR8 Myddosome signaling complex, where TIRAP contributes to Akt-kinase activation and the nuclear translocation of interferon regulatory factor 5 (IRF5). Recruitment of TIRAP to the TLR8 signaling complex promotes the expression and secretion of the IRF5-dependent cytokines IFNβ and IL-12p70 as well as, to a lesser degree, TNF. These findings reveal a new and unconventional role of TIRAP in innate immune defense.

## 1. Introduction

Toll-like receptors (TLRs) are one of the most studied groups of pattern-recognition receptors (PRRs), which recognize pathogen-associated molecular patterns and damage-associated molecular patterns. TLRs provide protection against both external and internal threats by initiating a pro-inflammatory response as well as activating and guiding the adaptive immune system to mount effector responses that will eliminate and/or ameliorate the problem [1]. Dysfunction or dysregulation of TLR responses can have dire consequences for the host, such as increased susceptibility to infection or excessive life-threatening inflammation. Thus, understanding how these receptors operate holds great potential for guiding preventive measures as well as the development of better treatments and new drugs for infectious and autoimmune diseases, cancer, and cardiovascular disease [2]. Through the years, some TLRs have received more attention than others, often due to the availability of easily accessible models and their convenience for study, but with technological and scientific advances, this gap is closing. TLR8 provides a classic example of a less-studied TLR as it is non-functional in rodents and, thus, application of murine models/KOs is limited [3].

TLR8 is expressed by several immune cells, in particular monocytes, macrophages, myeloid dendritic cells, and neutrophils [4,5,6,7]. Localized at the endosomes, TLR8 recognizes ribonuclease T2 degradation products of single-stranded RNA (ssRNA) of various origin: viral, bacterial, protozoan, and (possibly human) endogenous RNA [5,8,9,10,11,12]. With this repertoire of potential ligands, TLR8 might be relevant for the defense against a broad range of infections, as well as for driving autoimmune diseases [3,6,11]. TLR8 ligand-binding in the endosomal lumen induces conformational changes that result in the dimerization of the cytosolic TIR domains. This allows myeloid differentiation primary response gene 88 (MyD88) to bind, followed by the recruitment of interleukin-1 receptor-associated kinase 4 (IRAK4) and interleukin-1 receptor-associated kinase 1 (IRAK1), resulting in the formation of the active Myddosome complex. The signal is transduced via TNF-receptor-associated factor 6 (TRAF6) and transforming growth-factor-β-activated kinase 1 (TAK1), which activates downstream mitogen-activated protein kinase (MAPK) cascades and inhibitor of nuclear-factor kappa B kinase subunit beta (IKKβ), culminating in the activation of transcription factors such as nuclear-factor kappa B (NF-κB) and activator-protein 1 (AP-1) [1]. In human primary macrophages, TLR8-activated TAK1 also signals via IKKβ to induce the nuclear translocation of interferon regulatory factor 5 (IRF5), which is critical for the expression of interferon β (*IFNβ*) and interleukin-12 subunit alpha (*IL-12A*) genes [13]. 

The phosphoinositide 3-kinase (PI3K)-Akt serine/threonine kinase pathway is well-known to be involved in the regulation of metabolism and survival, and its dysregulation is closely linked to tumor development [14]. It has also been implicated in regulating TLR-mediated responses, although the reports are divergent, with evidence of both pro- and anti-inflammatory effects [15]. Aksoy et al., found that in human monocyte-derived DCs, PI3K negatively regulated the expression of *IFNβ* in TRIF-dependent signaling downstream of TLR3 and TLR4 [16], whereas Guiducci et al., found PI3K–Akt necessary for nuclear translocation of IRF7 and expression of TLR7 and TLR9-induced type I IFNs in human plasmacytoid predendritic cells [17]. These studies highlight the importance of taking cell-type specific differences into account when investigating the role of PI3K–Akt in TLR signaling.

TLR2 and TLR4 require a Toll-internleukin-1-receptor (TIR) domain containing adaptor protein/MyD88 adaptor-like (TIRAP) to attract MyD88 to the signaling complex. For some time, the restricted role of TIRAP for these plasma-membrane-localized TLRs seemed apparent, given the phosphatidylinositol-4,5-bisphosphate (PI(4,5)P2)-binding motif in the N-terminal domain of TIRAP, which attracts this adaptor protein to the plasma membrane where PI(4,5)P2 is abundant [18,19]. However, Bonham et al., have shown that TIRAP is also capable of binding other phosphoinositides, PI(3)P and PI(3,5)P2, on endosomal membranes, and mediates signaling from endosomal TLR7 and TLR9 in murine cells [20]. Recently, some evidence of TIRAP involvement in signaling from TLR7 and TLR9 in human cells was provided by Leszczynska et al., and Zyzak et al. [21,22]. In both studies, TIRAP is suggested to regulate *IFNβ* expression by regulating ERK1/2 (MAPK3/MAPK1) activation. Whether TIRAP could contribute to signaling from TLR8, the third member of the TLR7 subgroup of TLRs, has not yet been explored.

Here we show that TIRAP is regulating the expression and secretion of TLR8-induced IFNβ and IL-12A cytokines in human primary monocytes and monocyte-derived macrophages (MDMs). We propose that TIRAP is recruited to the activated Myddosome, from where it connects to Akt activation, contributing to the nuclear translocation of IRF5 and subsequent expression of *IFNβ* and *IL-12A* genes. In addition, TIRAP can enhance TLR8 signaling via the TAK1-pathway, thus modulating the expression of other cytokines such as TNF.

## 2. Materials and Methods

### 2.1. Cells and Reagents

Human buffy coats and serum were from the blood bank at St. Olavs Hospital (Trondheim, Norway), with approval by the Regional Committee for Medical and Health Research Ethics (REC) in Central Norway (no. 2009/2245). Primary human monocytes were isolated from the buffy coat by adherence, as previously described [23]. Monocytes were maintained in RPMI1640 (Sigma, Merck, Darmstadt, Germany), supplemented with 30% of pooled human serum. MDMs (used in the *TIRAP* silencing experiments) were obtained by differentiating cells for 12 days in RPMI1640 with 10% human serum and 20 ng/mL rhM-CSF (#216-MC-025, R&D Systems, Minneapolis, MN, USA). Ultrapure K12 LPS from *E. coli*, thiazoloquinoline compound CL075, and synthetic diacylated lipoprotein FSL-1 (Pam2CGDPKHPKSF) were from InvivoGen (San Diego, CA, USA). For stimulation of the primary cells, LPS and FSL-1 were used at concentration 100 ng/mL, CL075—2 µg/mL. IRAK4 inhibitor PF-06426779 (Merck, Darmstadt, Germany), selective allosteric pan-Akt inhibitors MK-2206 (#1032350-13-2, Axon Medchem, Groningen, Netherlands), Miransertib) and ATP-competitive pan-Akt inhibitor Capivasertib ((#1313881-70-7 and #1143532-39-1, MedChemExpress, Sollentuna, Sweden) were diluted in DMSO at concentration 5 mM and stored at −80 °C; working solutions were prepared in cell-culture media immediately before use. Preparation of THP-1 *TIRAP* KO cells using LentiCRISPRv2 plasmid [24] is described in Appendix A (available online).

### 2.2. Antibodies

The following primary antibodies were used: mouse GAPDH (ab9484), rabbit β-tubulin (ab6046) from Abcam (Cambridge, UK); rabbit phospho-Akt Ser473 (D9E XP), phospho-p38 MAPK (T180/Y182), phospho-STAT1 (Tyr701) (58D6), phospho-TAK1 (T184/187) (90C7), phospho-JNK (81E11) (T183/Y185), phospho-p44/42 MAPK (ERK1/2) (Thr202/Tyr204), IκBα (44D4), IRAK1 (D51G7), MyD88 (D80F5), Histone H3 (3H1), and phospho-NF-κB p65 (Ser536) (93H1) from Cell Signaling Technology (Danvers, MA, USA); rabbit PCNA Abs were from Santa Cruz Biotech (Santa Cruz, CA, USA); sheep IRF5 and IRAK4 were from MRC-PPU Reagents (University of Dundee, Dundee, UK); goat TIRAP polyclonal Abs were from Invitrogen (#PA5-18439, Waltham, MA, USA), and mouse STAT1 antibodies were from BD Biosciences (#610185, Wokingham, UK). Secondary antibodies (HRP-linked) were from DAKO Denmark A/S (Glostrup, Denmark).

### 2.3. siRNA Treatment

Oligos used for silencing were AllStars Negative Control siRNA (SI03650318) and FlexiTube Hs_TIRAP_10 siRNA (SI03075135) (QIAGEN, Germantown, AR, USA). PBMCs were seeded in 24-well plates (NUNC, ThermoFisher Scientific, Waltham, MA, USA), 1.5 × 10^6^ cells per well, and differentiated to MDMs as described above. On day 7 and day 9, cells were transfected by silencing and control oligo (20 nM final concentration) using Lipofectamine 3000 (Invitrogen, ThermoFisher Scientific, Waltham, MA, USA), as suggested by the manufacturer. Cells were stimulated by LPS, CL075 or FSL-1, or used for bacterial infections in 48 h after second transfection.

### 2.4. RT-qPCR

Total RNA was isolated from the cells using Qiazol reagent (QIAGEN, Germantown, AR, USA), and chloroform extraction was followed by purification on RNeasy Mini columns with DNAse digestion step (QIAGEN). cDNA was prepared with a Maxima First Strand cDNA Synthesis Kit for a quantitative real-time polymerase-chain reaction (RT-qPCR) (ThermoFisher Scientific, Waltham, MA, USA), in accordance with the protocol of the manufacturer, from 400–600 ng of total RNA per sample. Q-PCR was performed using the PerfeCTa qPCR FastMix (Quanta Biosciences, Gaithersburg, MD, USA) in replicates and cycled in a StepOnePlus™ Real-Time PCR cycler (ThermoFisher Scientific, Waltham, MA, USA). The following TaqMan^®^ Gene Expression Assays (Applied Biosystems^®^, ThermoFisher Scientific, Waltham, MA, USA) were used: *IFNβ* (Hs01077958_s1), *TNF* (Hs00174128_m1), *TBP* (Hs00427620_m1), *TIRAP* (Hs00364644_m1), *IL-6* (Hs00985639_m1), *IL-1β* (Hs01555410_m1), *IL-12A* (Hs01073447_m1), and *IL-12B* (Hs01011518_m1). The level of *TBP* mRNA was used for normalization and the results presented as a relative expression compared to the control’s untreated sample. Relative expression was calculated using Pfaffl’s mathematical model [25]. Graphs and statistical analyses were made with GraphPad Prism v9.1.2 (Dotmatics, Bishops Stortford, UK), with additional details provided in the figure legends and statistics paragraph (Section 2.11).

### 2.5. ELISA and BioPlex Assays

TNF level in supernatants of human macrophages was determined using human TNF-alpha DuoSet ELISA (DY210-05) (R&D Systems, Minneapolis, MN, USA), IFNβ level—using VeriKine-HSTM Human Interferon-Beta Serum ELISA Kit from PBL Assay Science (Piscataway, NJ, USA). Other cytokines (IL-12p70, IL-6, MCP-1, IL-8) were analyzed using BioPlex cytokine assays from Bio-Rad, in accordance with the instructions of the manufacturer, using the Bio-Plex Pro™ Reagent Kit III and Bio-Plex™ 200 System (Bio-Rad, Hercules, CA, USA).

### 2.6. Cell Fractionation

Stimulated human primary monocytes were detached by accutase treatment and collected by centrifugation. Cell pellets were washed by once by PBS with 2% FCS. For preparation of total lysate, a cell pellet from one of the wells per condition was collected and lysed by RIPA lysis buffer (150 mM NaCl, 50 mM TrisHCl (pH7.5), 1% Triton X100, 5 mM EDTA, protease inhibitors, phosphatase inhibitors). The remaining cells were resuspended in Buffer A (50 mM NaCl, 10 mM HEPES pH = 8, 500 mM sucrose, 1 mM EDTA, 0.2% Triton-X100), and the samples were vortexed and centrifuged (5000 rpm, 5 min, 4 °C). Cytosolic fraction in the supernatant was transferred to clean tubes. Pellets with nuclei were washed with Buffer B (50 mM NaCl, 10 mM HEPES (pH = 8), 25% glycerol, 0.1 mM EDTA), and centrifuged, the supernatants were discarded, and the pellets were resuspended in Buffer C (350 mM NaCl, 10 mM HEPES (pH = 8), 25% glycerol, 0.1 mM EDTA) with added Benzonaze endonuclease (Merck) and incubated on ice for 30 min, followed by centrifugation (15,000 rpm, 15 min, 4 °C) to extract nuclear proteins. 

### 2.7. Western Blotting

Cell lysates for pSTAT1 analysis were prepared by simultaneous extraction of proteins and total RNA using Qiazol reagent (QIAGEN, Germantown, AR, USA), as suggested by the manufacturer. Extracted total RNA was used for RT-qPCR, while protein samples were used for simultaneous analysis of protein expression/post-translational modifications. Protein pellets were dissolved by heating the samples for 10 min at 95 °C in a buffer containing 4 M urea, 1% SDS (Sigma, Merck, Darmstadt, Germany), and NuPAGE^®^ LDS Sample Buffer (4X) (Thermo Fisher Scientific, Waltham, MA, USA), with a final 25 mM DTT in the samples. Otherwise, lysates were made using 1X RIPA lysis buffer. For traditional Western blot analysis, we used pre-cast protein gels NuPAGE™ Novex™. Proteins were transferred to iBlot Transfer Stacks by using the iBlot Gel Transfer Device (ThermoFisher Scientific, Waltham, MA, USA). The blots were developed with the SuperSignal West Femto (ThermoFisher Scientific) and visualized with the LI-COR ODYSSEY Fc Imaging System (LI-COR Biotechnology, Lincoln, NE, USA). For densitometry analysis of the bands, Odyssey Image Studio 5.2 software (LI-COR Biotechnology, Lincoln, NE, USA) was used, and the relative numbers of bands’ intensity were normalized to the intensities of the respective loading-control protein (GAPDH, or PCNA, or β-tubulin). Loading-control-protein expression was always performed on the same membrane as the protein of interest.

### 2.8. Immunoprecipitations

PBMC-derived monocytes for endogenous IPs were lysed using 1 X lysis buffer (150 mM NaCl, 50 mM TrisHCl (pH 8.0), 1 mM EDTA, 1% NP40) and supplemented with EDTA-free Complete Mini protease Inhibitor Cocktail Tablets as well as a PhosSTOP phosphatase-inhibitor cocktail from Roche, with 50 mM NaF and 2 mM Na_3_VO_3_ (Sigma, Merck, Darmstadt, Germany). Immunoprecipitations (IPs) were carried out on rotator at +4 °C for 4 h by co-incubation of the lysates from the stimulated cells (400–500 μg of protein/IP) with specific anti-TIRAP antibodies covalently coupled to Dynabeads (M-270 Epoxy, Thermo Fisher Scientific, Waltham, MA, USA), as suggested by the manufacturer, followed by extensive washing of the beads in a lysis buffer. Co-precipitated complexes were eluted by heating the samples in a 1× loading buffer (LDS, Invitrogen), without reducing the reagent to minimize the antibodies’ leakage to the eluates. Eluates were transferred to clean tubes, followed by the addition of DTT to the 40 mM concentration, heating, and Western blot analysis. 

### 2.9. Immunofluorescence and Scan^R Analysis

Monocytes were isolated from PBMC using CD14 MicroBeads UltraPure (Miltenyi Biotech) and seeded in 96-well glass-bottom plates (P96-1.5H-N, Cellvis, CA, USA) at 50 K cells/well and in 24-well culture plates (250 K cells/well). Monocytes were differentiated into MDMs and transfected two times with a *TIRAP* siRNA and AllStar siRNA control, using the standardized protocol. MDMs were left untreated or stimulated with CL075 (Invivogen) at 2 μg/mL for 60 min, with four technical replicates per condition. Fixation, immunostaining with anti-human IRF5 mAb (Abcam, #10T1) and anti-human p65 XP mAb (Cell Signaling Technology, #8242, Danvers, MA, USA), and Scan^R high-throughput imaging (Olympus Europa SE & Co, Hamburg, Germany) were done, as previously described in detail [13]. Quantification of IRF5 nuclear translocation was done with Scan^R analysis software (v2.8.1) and calculated as a percentage of the positively stained nuclei multiplied by the mean fluorescence-intensity value (MFI) of the positively stained nuclei. Silencing efficiency of the TIRAP gene for each donor (*n* = 6) was examined by RT-qPCR, using the parallel 24-well plates.

### 2.10. Bacteria and Infection Experiments

Anonymized clinical isolates of GBS, *S. aureus*, and *E. coli* were from a diagnostic collection by the Department of Medical Microbiology, St. Olavs Hospital, Trondheim, Norway. For infection experiments, blood-agar colonies were picked and grown in Todd-Hewitt Broth (GBS) or Tryptic Soy Broth (*E. coli* and *S. aureus*), with vigorous shaking at 37 °C overnight. The bacteria cultures were diluted to the desired density (CFU/mL), based on OD600 measurements and calculations, as previously described in detail [26]. Cultures of MDMs in 24-well plates, with *TIRAP* siRNA or control siRNA pre-treatment, were incubated with bacteria at 37 °C for 60 min, before the killing of all extracellular bacteria with 100 μg/mL gentamicin. The MDM cultures were incubated further for a total challenge time of four hours. Cell lysis and RNA purification was done with the RNeasy 96 Plus kit (QIAGEN), followed by cDNA synthesis with Maxima cDNA synthesis kit (Thermo Fisher Scientific), and qPCR analysis of the cytokine expression. 

### 2.11. Statistical Analysis

Data that were assumed to follow a log-normal distribution was log-transformed prior to statistical analysis. RT-qPCR was log-transformed and analyzed by Repeated Measurements Analysis of Variance (RM-ANOVA), or a mixed model if there was missing data, followed by Holm-Šídák’s multiple comparisons post-test. Scan^R data were log-transformed and analyzed with a paired *t*-test (two-sided). ELISA data was analyzed using a Wilcoxon matched-pairs signed-rank test. All graphs and analyses were generated with GraphPad Prism v9.1.2 (Dotmatics, Bishops Stortford, UK). 

## 3. Results

### 3.1. TIRAP-Silencing Decreases TLR8-Mediated Cytokine Expression and Secretion in Human Primary MDMs

Based on the reported links between endosomal TLRs and TIRAP expression [20,21,22], we have questioned whether TIRAP could also be involved in the regulation of signaling via TLR8 in human immune cells. To address this, we performed *TIRAP* silencing in human primary MDMs prior to stimulation with the thiazoquinoline compound CL075, which is a synthetic ligand specific for TLR8 in human monocytes and MDMs [27]. Analysis of cytokine mRNA expression following TLR8 stimulation in *TIRAP*-silenced cells showed a significant reduction in *IFNβ* and *IL-12A* mRNA expression, with the most significant effect on *IL-12A* expression (Figure 1). Of the pro-inflammatory cytokines assessed following TLR8 stimulation, *TIRAP* silencing only affected *TNF* mRNA expression, both at two hours and four hours (Figure 1). TLR2 and TLR4 stimulation was conducted in parallel, since TIRAP is an important bridging adaptor for these TLRs [18,28]. Indeed, *TIRAP* silencing led to decreased *TNF*, *IL-6*, *IL-1β*, *IL-12A*, and *IL-12B* mRNA expression following TLR2 and TLR4 stimulation (Appendix A). TLR2 did not induce *IFNβ* mRNA expression or secretion in human MDMs, in agreement with earlier studies [29]. Notably, in the case of TLR2 and TLR4 stimulation, we have not observed a significant effect on the early induction of pro-inflammatory cytokines (Appendix A), which could be due to the still-sufficient amount of residual TIRAP protein for the initiation of signaling, despite a marked decrease in TIRAP-protein expression in silenced MDMs (Appendix A).

To avoid problems with inefficient silencing, knockout-model (KO) systems are widely used. We prepared *TIRAP* KO THP-1 human monocyte/macrophage-like subline by Crispr/Cas9 gene editing, combined with subsequent TLR8 overexpression to achieve more robust type I IFNs induction in these cells. However, there was a noticeable change in TLR4 signaling in *TIRAP* KO THP-1 cells from two weeks (early) to four weeks (late) of cultivation in puromycin-selection medium (Appendix A). TLR4-mediated *IFNβ* expression was inhibited in early *TIRAP* KO cells, while the effect was lost after prolonged cultivation (Appendix A). The response to LPS in *TIRAP*-silenced THP-1 cells was more comparable to early *TIRAP* KO cells (Appendix A). This instability of the KO cells could be due to compensatory mechanisms, and we, thus, considered silencing of the *TIRAP* gene as a better approach for pinpointing the impact of TIRAP in TLR signaling.

TLR2 and TLR4 induce the expression of pro-inflammatory cytokines via formation of the TIRAP/MyD88 complex [30]. Even though TLR4-dependent expression of *IFNβ* relies on TRAM/TRIF signaling from the endosomes, we observed that *TIRAP* silencing significantly reduced LPS-mediated *IFNβ* expression both in THP-1 cells and in primary human MDMs (Appendix A) [31].

To follow the protein levels of the secreted cytokines, we performed ELISA or Multiplex assays using supernatants from cells stimulated by TLR ligands for four hours. Indeed, *TIRAP* silencing reduced TLR4- and TLR8-mediated IFNβ and IL-12p70 secretion (Figure 2), and it also decreased the phosphorylation of STAT1 (Appendix A), a transcription factor in IFN-α/β-receptor (IFNAR) signaling that may be used as a surrogate readout for IFNβ secretion [32]. These results are corroborating the mRNA data, suggestive of TIRAP involvement in regulation of IFNβ expression and secretion at later stages of signaling (2–4 h). TLR8-mediated *TNF* mRNA expression was also reduced upon *TIRAP* knockout (Figure 1), yet the level of secreted TNF was not significantly affected (Figure 2). Overall, our results indicate that TIRAP regulates TLR8-mediated signaling in human primary MDMs, with the strongest effect on expression and secretion of IRF5-regulated cytokines IFNβ and IL-12p70, and a less clear effect on the expression and secretion of proinflammatory cytokines.

### 3.2. TIRAP Silencing Inhibits TLR8-Dependent IL-12A Expression in Response to Bacterial Infection 

TLR8 can sense ssRNA of bacterial origin [5,9,26]. Given how common bacterial infections are, and the potential serious consequences associated with them, it is essential to understand the mechanistic interactions between the pathogen and immune cells. Group B streptococcus (*S. agalactiae*, GBS) and *S. aureus* are commensal bacteria but hold great invasive potential and can cause serious infections [33,34,35]. TLR8 was recently shown to be involved in the innate immune responses to these Gram-positive bacteria but had less impact on responses to Gram-negative bacteria [13,26,27]. As such, we wanted to address the importance of TIRAP in the TLR8-mediated responses to GBS and *S. aureus*. *TIRAP*-silenced primary MDMs were stimulated with TLR4 ligand LPS or TLR8 ligand CL075, or infected with clinical isolates of *E. coli*, GBS, or *S. aureus*, for a total of four hours and at two bacterial doses. Expression of *IFNβ*, *TNF*, *IL-12A*, *IL-12B*, and *IL-6* mRNA was analyzed by RT-qPCR (Figure 3). In accordance with our previous findings, successful *TIRAP* silencing reduced the expression of these cytokines following LPS treatment and infection by *E. coli* (Figure 3). *TIRAP* silencing also significantly reduced *IL-12A* mRNA expression following GBS infection, with a similar tendency for *IFNβ* induction, although statistical support was not achieved (Figure 3). These results indicated a clear effect of *TIRAP* silencing on TLR8-mediated *IL-12A* expression, with a similar yet non-significant trend for *IFNβ* mRNA (Figure 1 and Figure 3). In *S. aureus*-infected cells, *TIRAP* silencing significantly reduced *IL-6* and *TNF* mRNA expression, which could reflect the inhibition of TLR2-mediated pro-inflammatory signaling in *TIRAP*-silenced cells (Figure 3). We observed non-significant trends in the reduction in *IFNβ*, *IL-12A*, or *IL-12B* mRNA expression, following an *S. aureus* challenge (Figure 3). *S. aureus* is sensed both by TLR2 and TLR8, and TLR2 activation can suppress TLR8-IRF5-mediated induction of *IFNβ* and *IL-12A* [13], and it is possible that the combined TLR2 and TLR8 activation can diminish the requirement for TIRAP in the TLR8 mediated responses to *S. aureus*. Additional sensing mechanisms of whole live bacteria are also involved, such as complement and Fc receptors. These might compensate for the loss of TIRAP in our experiments, thus explaining the relatively small effects upon a bacterial challenge compared to pure TLR8-agonist stimulation. Thus, a kinetic study would be required to further clarify the impact of TIRAP in the sensing of live bacteria. Moreover, gene silencing has limitations, since it does not completely abrogate the expression or protein level of the silenced gene (Appendix A). Altogether, these results indicate that TIRAP can influence IRF5-mediated TLR8 signaling also during bacterial infection, which is especially clear for the induction of *IL-12A* expression by GBS.

### 3.3. TLR8-Mediated Nuclear Translocation of IRF5 Is Reduced by TIRAP Silencing

Based on our findings, which show that *TIRAP* silencing had the most prominent inhibitory effect on the expression and secretion of the IRF5-regulated cytokines IFNβ and IL-12p70 (Figure 1, Figure 2 and Figure 3), we addressed TLR8-mediated nuclear translocation of the transcription factors NF-κB p65/RelA (p65) and IRF5 after the stimulation of cells by the CL075 ligand. NF-κB, a heterodimer with p65 as one of the components of the dimer, mainly induces the expression of pro-inflammatory cytokines, such as *TNF* and *IL-6* [1,36]. In human monocytes and MDMs, TLR8 induces expression of *IFNβ* and *IL-12A* genes in an IRF5-dependent manner, while TNF is less affected by *IRF5* silencing [13,26,27].

MDMs were treated with *TIRAP* siRNA or control siRNA prior to stimulation with the TLR8 ligand CL075, and silencing efficacy was confirmed by RT-qPCR from one of the parallel wells (Figure 4a). NF-κB p65 and IRF5 were stained by specific antibodies in TLR8-stimulated (60 min) cells, and nuclear translocation was assessed by automated high-throughput fluorescence imaging and quantification with Scan^R (Figure 4). *TIRAP* silencing significantly attenuated the nuclear translocation of IRF5 after TLR8 stimulation, while p65 translocation was unaffected (Figure 4b,c).

Phosphorylation of IRFs and other transcription factors typically reflect their active state and is linked to their nuclear translocation and the activation of transcription of target genes [37,38]. To evaluate the potential regulation of IRF5 phosphorylation by TIRAP, we performed electrophoresis using Phos-tag gel [39] (Appendix A), since efficient phospho-IRF5 antibodies are not commercially available. Lysates of LPS-stimulated cells were included as a negative control since TLR4 signaling does not trigger IRF5 translocation in human MDMs, but rather it activates the IRF3-transcriptional factor in a TRAM/TRIF-dependent manner [13,26,40]. No clear alterations in the phosphorylation pattern of IRF5 or p65 in *TIRAP*-silenced cells after CL075 stimulation were revealed (Appendix A). Thus, *TIRAP* expression positively regulated the nuclear translocation of IRF5 downstream to TLR8 (Figure 4), while having no effect on p65 phosphorylation or nuclear translocation (Appendix A).

### 3.4. TIRAP Co-Precipitates with the Myddosome Complex Induced by TLR8 Dimerisation

TIRAP interacts with activated TLR2 and TLR4 dimers and recruits the MyD88 signaling adaptor, allowing formation of the Myddosome complex [18]. The role of TIRAP in the TLR8-MyD88 complex has not previously been addressed, and there are no published data on possible TIRAP recruitment to the TLR8-activated Myddosome. To address the recruitment of TIRAP to the TLR8-Myddosome complex, we performed immunoprecipitations with TIRAP-specific antibody-coated beads and lysates from human primary MDMs, stimulated with LPS (positive control), or CL075 (Figure 5). Indeed, TIRAP co-precipitated with MyD88, IRAK4, and IRAK1, the core signaling proteins of the Myddosome complex [1], not only in LPS-stimulated cells, but also upon stimulation via TLR8 (Figure 5).

To validate the IRAK4 band in the TIRAP precipitates (due to the IRAK4 size of 50–52 kDa, which is close to the size of IgG heavy chains), we examined TIRAP co-precipitations with IRAKs and MyD88 from LPS-stimulated THP-1 cells, using an IRAK4 inhibitor (PF-06426779) that induces a band size shift of IRAK4 due to the inhibition of IRAK4 autophosphorylation (Appendix A). Pre-treatment with the IRAK4 inhibitor decreased LPS-mediated IRAK1 posttranslational modifications and TAK1 phosphorylation in the lysates, resulting in the expected IRAK4 band-size shift in the precipitates (Appendix A). This shows that IRAK4 staining is specific for the chosen IPs conditions (Figure 5).

The shared time point of 15 min for the LPS and CL075-stimulated cells demonstrates that TIRAP recruitment to IRAKs and MyD88 was delayed for CL075-stimulated cells when compared with LPS-stimulated cells (Figure 5). Overall, the extent of IRAK1 modification in 15 min of TLR stimulation in MDMs showed great donor variation (not shown), as expected. IP results with lysates from a donor with a fast and strong response to CL075 were selected to demonstrate that even with fast IRAK1 activation (already within 15 min), TIRAP was not co-precipitating with the Myddosome-complex molecules (Figure 5). In contrast, TIRAP was recruited to TLR4-activated Myddosome within 15 min, even though this experiment revealed only weak IRAK1 modification following LPS stimulation (Figure 5). These observations are in line with the well-established role of TIRAP as adaptor that connects MyD88 to TLR4 [28,41]. 

Thus, TIRAP recruitment to the Myddosome complex was delayed and more prominent at 30–60 min after CL075 stimulation, when compared to LPS stimulation (Figure 5). Overall, we suggest that recruitment of TIRAP to TLR8 occurs after the Myddosome formation is initiated. This is in line with the concept of the direct interaction of endosomal TLRs with the signaling adaptor MyD88 [28,41]. TIRAP may, thus, not be required for connecting MyD88 to TLR8 to initiate the signaling but is rather recruited to the activated TLR8-signaling complex, and, thus, subsequently regulate downstream signaling. Overall, our data show that TIRAP is attracted to the TLR8 Myddosome and further support the hypothesis that TIRAP is involved in the regulation of TLR8 signaling. 

### 3.5. TLR8-Mediated Akt Phosphorylation Is Negatively Affected by TIRAP Silencing

To gain further insight into the role of TIRAP downstream of the TLR4 and TLR8 Myddosomes, we analyzed the phosphorylation/activation state of signaling intermediates in *TIRAP*-silenced human MDMs (Figure 6). *TIRAP* silencing was expected to have an inhibitory effect on the activation/phosphorylation of TLR4-mediated MyD88-dependent signaling molecules. Upon TLR ligation, MyD88 is recruited to TLR4 in a TIRAP-dependent manner [28], and IRAK4 kinase is subsequently attracted to MyD88 via death domain (DD) interactions, followed by IRAK1 recruitment and activation. As a result, IRAK1 is phosphorylated and poly-ubiquitinated, which induces the shift of IRAK1 band size from 80 kDa to 100 kDa or a significant reduction in the 80 kDa band [42,43,44]. Active IRAK1 forms the complex as well as promotes the phosphorylation and activation of TAK1-mitogen-activated kinase kinase kinase (MAPKKK) that acts upstream and induces ERK1/2, JNK1/2, and p38 MAPK phosphorylation and activation, while TAK1 also activates the canonical IKK complex (reviewed in [45]). IKKβ is crucial for IκBα (nuclear factor of kappa light-polypeptide gene enhancer in the B-cells inhibitor, α) phosphorylation, which leads to its degradation that is required for the activation of NF-κB (reviewed in [46]). IKKβ is also critical for the activation of IRF5 in TLR8 signaling [13,37]. Both p38MAPK and JNK1/2 positively regulate the transcriptional activity of the AP-1 (ATF-2-c-jun) transcriptional complex [47], which together with IRFs (IRF3 for TLR4 and IRF5 for TLR8) and NF-kB translocate to the nucleus and activate type I IFN promoters [13,40].

As could be seen from Figure 6a,c, in cells stimulated with LPS for 15–30 min, *TIRAP* silencing reduced the phosphorylation of TAK1, ERK1/2, JNK1/2, and p38 MAPK as well as the post-translational modification of IRAK1 (100 kDa band), and resulted in less effective degradation of IκBα. Silencing of *TIRAP* had some inhibitory effect on TLR8-mediated TAK1 and p38 MAPK phosphorylation, while not affecting IRAK1 posttranslational modifications or the degradation of IκBα (Figure 6a,b). Indeed, de-phosphorylation of TAK1 in *TIRAP*-silenced cells was faster when compared to control cells upon CL075 stimulation (Figure 6a,b), which may indicate a possible role of TIRAP in the stabilization of IRAK1/TABs/TAK1 signaling complex and increased TAK1-mediated cytokine production.

PI3Ks and its downstream target, serine/threonine-kinase Akt (PKB), is activated by many receptors, including TLRs, and is known to regulate macrophage survival and migration as well as the response to different metabolic and inflammatory signals in macrophages [48,49,50]. Phosphorylation of Akt (serine 473, S473) reflects a fully activated Akt kinase [51]. The most consistent effect of *TIRAP* silencing upon the ligation of TLR8 across PBMCs from several donors was the decreased phosphorylation of Akt, particularly 45–60 min after TLR8 activation (Figure 6a,b), which correlates with the timeframe when TIRAP is recruited to the TLR8-induced Myddosome (Figure 5). These results indicate that TLR8 signaling may be coupled to the PI3K/Akt pathway, and that TIRAP positively regulates TLR8-mediated activation of Akt (Figure 6a,b). In comparison, in LPS-stimulated cells, *TIRAP* silencing had not much effect on the phosphorylation of Akt (Figure 6a,c).

Previously, Guiducci et al., reported that TLR7 stimulation induces phosphorylation of Akt, and inhibition of Akt reduces nuclear translocation of IRF7 and type I IFNs’ induction [17]. Thus, positive regulation of TLR8-mediated IRF5 nuclear translocation by TIRAP (Figure 4) might be mechanistically linked to the regulation of Akt (Figure 6a,b).

### 3.6. p38 MAPK Inhibition Is Not Affecting TLR8-Mediated IRF5 and p65 Nuclear Translocation

*TIRAP* silencing resulted in reduced p38 MAPK phosphorylation, upon stimulation by TLR8 ligand (Figure 6). To investigate the role of p38 MAPK in TLR8 signaling, we pre-treated monocytes with a selective p38 MAPK inhibitor BIRB 796 and several other control inhibitors, followed by the stimulation of cells with CL075 (1 µg/mL) for one and two hours Overall, inhibition of p38 MAPK had no effect on nuclear translocation of IRF5 or p65 (Appendix A). In contrast, TAK1 kinase inhibitor (5z-7-oxozeaenol) blocked IRF5 translocation, but not p65 translocation, while inhibiting IKKβ with a IKKII–VIII inhibitor that blocked both p65 and IRF5 translocation, as shown in our earlier study [13]. These data suggest that decreased phosphorylation of p38 MAPK does not explain the reduction in IRF5 nuclear translocation upon *TIRAP* silencing. Still, inhibition of p38 MAPK strongly inhibited the expression of *IFNβ* and *TNF* mRNA in 2 h of CL075 stimulation, similar to TAK1 inhibition (Appendix A). Since p65 nuclear translocation was not affected by the p38 MAPK inhibitor, this could be explained by attenuation of the AP-1 transcriptional complex activity in BIRB796-treated cells, which would result in reduced cytokine induction according to the established role for AP-1 in cytokines’ promoter activity [47]. Overall, we conclude that TIRAP may regulate TLR8 signaling via two distinct pathways: an Akt pathway and the TAK1 pathway that enhances p38 MAPK activation. Both pathways contribute to cytokine induction. The regulation of *IFNβ* and *IL-12A* expression by modulation of nuclear IRF5 levels is the most marked effect of *TIRAP* silencing, which could not be explained by the decreased p38 MAPK activation in silenced cells. Thus, we proceeded with testing the effect of Akt inhibition on TLR8-mediated *IFNβ* and *IL-12A* expression as well as IRF5 nuclear translocation.

### 3.7. Akt Inhibition Decreases TLR8-Mediated Expression of IFNβ and IL-12A Genes

To further examine the role of Akt in TLR8 signaling, and particularly in the regulation of *IFNβ* and *IL-12A* expression, we used specific Akt inhibitors. Two allosteric inhibitors (MK-2206 and Miransertib) and one ATP-competitive Akt inhibitor (Capivasertib) had a similar inhibitory effect on the TLR8-mediated expression of *IFNβ* and *IL-12A* (Appendix A). We proceeded with the inhibitor MK-2206 and pre-treated primary human monocytes prior to stimulation with CL075 (Figure 7). Indeed, Akt inhibition resulted in a significant decrease in *IFNβ* and *IL-12A* mRNA in two hours of TLR8 stimulation, with no effect on *TNF* mRNA expression (Figure 7). Of the TLR4-mediated responses, Akt inhibition resulted in the reduced *TNF* expression, without effect on *IFNβ* and *IL-12A* expression, suggesting a different mechanism for the regulation of TLR4-mediated *IFNβ* expression by TIRAP. 

Due to the previously detected link between activation of TAK1 and IRF5, we addressed the possible impact of Akt inhibition on the phosphorylation of TAK1 S172 and the downstream phosphorylation of p38 MAPK (Appendix A). Akt inhibition was efficient, since the allosteric Akt inhibitor blocked S473 phosphorylation, as previously revealed [14]. Interestingly, Akt inhibition rather increased the phosphorylation of TAK1 and downstream p38 MAPK after both TLR4 and TLR8 stimulation (Appendix A). At the same time, Akt inhibition by MK-2206 reduced the TLR8-mediated phosphorylation of STAT1 (Appendix A), which correlates with *IFNβ* gene expression (Figure 7). Overall, these results suggest that Akt is involved in the positive regulation of TLR8 signaling, leading to the expression of *IFNβ* and *IL-12A*, although mechanistically it appears not to be mediated by increased TAK-1 activation. In contrast to TLR8, inhibition of Akt had no significant effect on TLR4-mediated *IFNβ* and *IL-12A* induction.

### 3.8. Akt Inhibition Decreases TLR8-Mediated Nuclear Translocation of IRF5 

As we have shown, nuclear translocation of IRF5 in TLR8-stimulated *TIRAP* silenced cells was significantly reduced without a clear effect on IRF5 phosphorylation (Figure 4). We, thus, investigated the effect of Akt activity on nuclear translocation and phosphorylation of IRF5 in MDMs using subcellular fractionation (Figure 8). LPS-stimulated cells were included as a negative control for IRF5 nuclear translocation, and anti-phospho-Akt (S473) was used to demonstrate an efficient Akt blockade. Histone 3 levels were analyzed for the normalization of nuclear extracts, while GAPDH used to control for the possible contamination of nuclear extracts with cytosolic content, which was not the case (Figure 8). Overall, inhibition of Akt markedly reduced IRF5 nuclear translocation in 60 min of TLR8 stimulation (Figure 8). As with *TIRAP* silencing, the total phosphorylation pattern of IRF5 in total lysates of monocytes was not altered upon Akt inhibition (Appendix A). Together, these data suggest TIRAP is involved in a crosstalk between TLR8 and Akt, which contributes to IRF5 nuclear translocation and the expression of *IFNβ* and *IL-12A* genes (Figure 9).

## 4. Discussion

TIRAP/Mal is a critical bridging adaptor that connects MyD88 to TLR2 and TLR4 at the plasma membrane. However, it is now clear that the role of TIRAP in TLR signaling is much more complex (reviewed in [18]), with even some TLR-independent functions discovered for TIRAP (reviewed in [52]).

In its N-terminal part, TIRAP contains a phosphoinositide (PI)-binding domain (PBD), which interacts with phosphatidylinositol 4,5-bisphosphate (PtdIns(4,5)P2)-enriched membranes [19,20,53]. Bonham et al., demonstrated that TIRAP PBD is also capable of binding PtdIns(3)P on endosomal membranes, and when murine-bone-marrow-derived macrophages (BMDMs) are challenged with natural ligands (influenzas virus and Herpes simplex virus), TIRAP regulates signaling via TLR7 and TLR9, with a particular impact on IFNα expression [20].

Here, we show that TLR8 also utilizes TIRAP in its IRF5-dependent signaling pathway in human primary monocytes and MDMs, which has not previously been reported. Moreover, our findings suggest that TIRAP plays an unconventional role in TLR8 signaling and most likely is recruited after the formation of the proximal TLR8-Myddosome complex, subsequently enhancing Akt/PKB activation (Figure 9).

To investigate the potential involvement of TIRAP in TLR8 signaling, we based our study on *TIRAP* silencing in human primary phagocytes. Despite the quite high variability in kinetics and magnitude of TLR signaling in the human primary cells from healthy human subjects, which can be of genetic as well as non-genetic causes, our experiments provide some important advantages over studies with cell lines. The response of primary cells more accurately reflects the natural human-cell biology and host–pathogen interactions, so it is, therefore, of higher relevance.

Our data on the mRNA expression and cytokine secretion of IFNβ and IL-12A in *TIRAP*-silenced cells showed a significant reduction in these responses upon TLR8 stimulation, while the effect on pro-inflammatory cytokines was not as clear. We have recently demonstrated that TLR8 is a dominant TLR in the response to the Gram-positive bacteria, such as GBS and *S. aureus*, in human primary monocytes and MDMs [13,26,27]. We, therefore, addressed the contribution of TIRAP in the response to Gram-positive and Gram-negative bacterial infections in MDMs and revealed a partial dependency of TIRAP in the regulation of cytokine production with all the examined bacteria. The impact of TIRAP in *E. coli*-induced cytokine production could be mainly attributed to LPS-activated TLR4-signaling, while a partial reduction in TNF and IL-6 production induced by *S. aureus* may involve TLR2 signaling. Genetic *TIRAP* deficiency in humans impairs both TLR4 and TLR2 signaling, though the TLR2-response in macrophages and susceptibility to *S. aureus* infections can be rescued in vivo by lipoteichoic acid (LTA)-specific IgG antibodies, likely due to a compensatory mechanism via Fc-receptor (CD32) engagement [54]. We have shown earlier that the GBS-induced cytokine production in myeloid cell cultures is almost entirely TLR8-mediated [27], and the significant reduction in *IL-12A* expression, thus, demonstrates a role of TLR8–TIRAP signaling during a challenge with a viable Gram-positive bacterium. Even though induction of *IL-12A* and *IFNβ* by the Gram-positive bacteria is mainly TLR8 dependent [27], our data only revealed a tendency of attenuated *IFNβ* expression after *TIRAP* silencing, which did not reach statistical support, possibly due to underpowered statistics. It is also possible that the selected time point for gene-expression analysis was sub-optimal, or that the triggering of several signaling mechanism by the whole live bacteria (e.g., TLR2, TLR8, Fc-receptors, complement receptors, etc.) could compensate for the reduced TIRAP levels. Moreover, considerable levels of TIRAP protein remain in the cells after *TIRAP* silencing. The effect of *TIRAP* silencing was more prominent for TLR2 and TLR4 signaling with prolonged stimulation, which may imply that the levels of TIRAP in silenced cells are sufficient to initiate proximal signaling but not to sustain the cytokine production. Generation of *TIRAP* KO THP-1 cells was done as an alternative strategy, but the phenotype appeared unstable, possibly due to compensatory signaling mechanisms. Thus, the real contribution of TIRAP to TLR8-mediated cytokine induction might be more prominent, both for purified agonists and whole bacteria, and this issue, thus, warrants further studies.

As we have already noted, TIRAP appears to be more important for the expression of IRF5-dependent genes *IFNβ* and *IL-12A* than for TLR8-regulated pro-inflammatory cytokines. A similar differential regulation of *IFNβ* by TIRAP was also observed by Zyzak et al., in TLR9 signaling in human PBMCs and the microglia cell line [22], and by Lesczcynska et al., in TLR7 signaling in the human dendritic cell line [21]. Both reports implicate ERK1/2 in the TIRAP-dependent effects observed. Similar to our findings, Leszczynska et al., conclude that the TIRAP-dependent effects on *IFNβ* expression are mediated by IRF7, whereas Zyzak et al., link the non-canonical NF-kB pathway to IFNs-expression regulation [21,22]. However, especially in the latter report [22], most of the research into the mechanisms shown are done in murine cells.

TLR8-mediated *IFNβ* and *IL-12A* gene expression in human primary monocytes and macrophages is dependent on activation of IRF5 [13,27]. Our findings provide evidence that TIRAP is involved in the TLR8-IRF5 signaling mechanism. In contrast to the findings by Zyzak et al. [22] and Leszczynska et al. [21] regarding TLR9 and TLR7, *TIRAP* silencing in human MDMs did not alter ERK1/2 activation but, consistently, reduced the phosphorylation of Akt kinase. The PI3K–Akt pathway can regulate cellular metabolism and survival, and its dysregulation is firmly linked to tumor development [14]. The PI3K–Akt pathway is also implicated in the regulation of TLR signaling, with evidence of both pro- and anti-inflammatory effects [15,50,55,56]. Guiducci et al., showed that nuclear translocation of IRF7 and type I IFNs expression is enhanced by PI3K–Akt signaling, following TLR7 and TLR9 activation in human pDCs [17]. Lima et al., reported crosstalk between TLR9 and PI3Kγ in human PBMCs [57], while Sarkar et al., demonstrated that TLR3-dependent activation of the PI3K–Akt axis induces IRF3 phosphorylation and nuclear translocation in HEK293 cells [58]. Similar to these findings, here we show that Akt is involved in the regulation of *IFNβ* and *IL-12A* induction upon TLR8-IRF5 signaling. However, Akt could not be linked to TLR4-induced *IFNβ* and *IL-12A* expression in human MDMs.

Lopez-Pelaez et al., previously identified that IKKβ phosphorylates Ser462 in IRF5 to induce nuclear translocation and subsequent expression of *IFNβ* in a TLR7-stimulated human pDC cell line [37]. We find that Akt inhibition and *TIRAP*-silencing reduced IRF5 nuclear translocation upon TLR8 stimulation, though IRF5 phosphorylation appeared unaffected. However, it is possible that changes in the phosphorylation of the specific sites in IRF5 may not be detected by the analysis of the total phosphorylation pattern of IRF5.

Overall, we found that *TIRAP* silencing most consistently decreased the phosphorylation of Akt in human MDMs, and that inhibition of Akt had a comparable effect with *TIRAP* silencing. It might be possible that recruitment of TIRAP followed by Akt activation is adding another layer to the regulation of IRF5 activation and IRF5-dependent gene expression, either by direct phosphorylation of IRF5, or by regulating the activity of transport proteins involved in IRF5 nuclear translocation. Recently, a TLR adaptor interacting with SLC15A4 on the lysosome (TASL) was identified as a critical endosomal adapter for IRF5 activation in TLR7-9 signaling in human cells [59], and further studies are necessary to deduce the precise signaling events upstream of IRF5 activation as well as the specific roles of TIRAP and Akt in these pathways.

As an endosomal ssRNA-sensing receptor, TLR8 is relevant both for viral and bacterial infections [5,9,13,26,27,60,61,62]. IFNs are particularly important during viral infections, as they induce the expression of gene-encoding proteins with anti-viral effects, such as inhibiting the viral replication, assembly, and release of the virus particle [1]. Thus, further exploration of the role of TIRAP in virus-induced TLR8 signaling (such as Influenza A Virus, HIV, West Nile Virus [61]) is warranted. Cell-type specific differences in the expression level and utilization of TIRAP in TLR8 signaling should also be addressed. Furthermore, SNPs in TIRAP are associated to the incidence and severity of several diseases, such as tuberculosis, HIV, and systemic lupus erythematosus (SLE) [18] as well as infections/diseases in which TLR8 is likely to play a role [63,64,65]. Thus, understanding the contribution of TIRAP to the fine-tuning of TLR8 signaling might be of significant clinical relevance. 

## Figures and Tables

**Figure 1 biomedicines-10-01476-f001:**
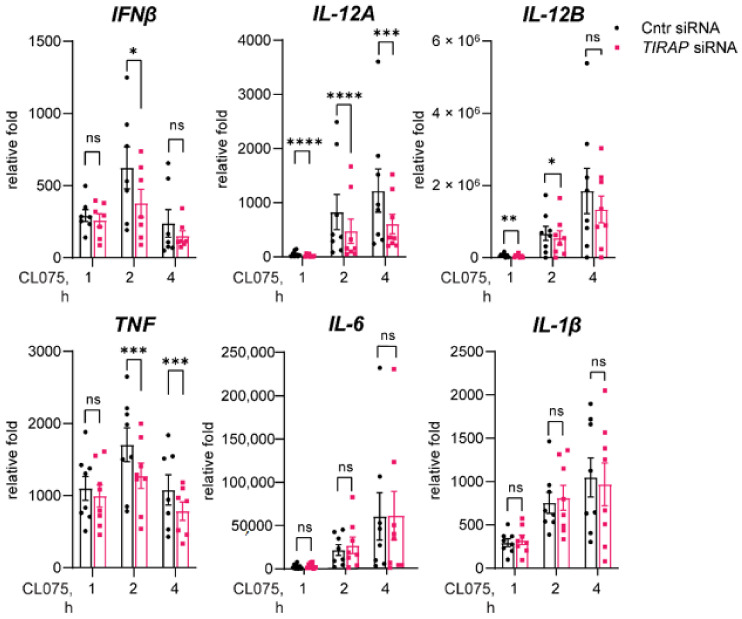
*TIRAP* silencing in primary human MDMs significantly decreases TLR8-mediated *IFNβ* and *IL-12A* expression. Macrophages were transfected with control or *TIRAP*-silencing oligo, followed by stimulation with TLR8 ligand CL075 (2 μg/mL) for the indicated time. RT-qPCR analysis of cytokine-gene expression after stimulation by CL075 in consecutive experiments with cells from different donors (*n* = 6–8). Data for cytokine expression induced by CL075 stimulation were normalized to untreated sample and presented as a mean relative fold change +SEM. Statistical testing was done by 2-way RM-ANOVA including a post-test, as described (* *p* < 0.05, ** *p* < 0.01, *** *p* < 0.001, **** *p* < 0.0001, and ns—non-significant).

**Figure 2 biomedicines-10-01476-f002:**
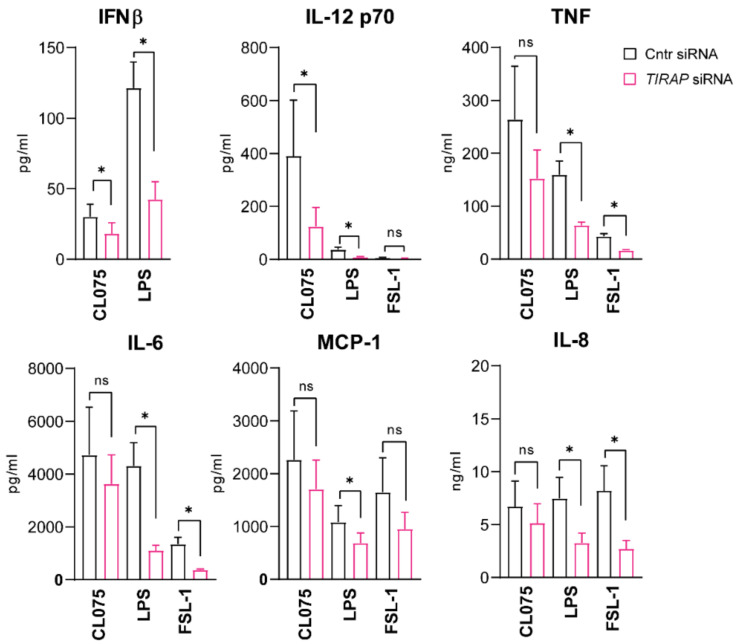
*TIRAP* silencing significantly inhibits TLR8-mediated IFNβ and IL12 p70 secretion by primary human macrophages. IFNβ and TNF secretion in 6–8 consecutive experiments with cells from different donors were analyzed by specific ELISA kits, with other cytokines’ secretion addressed by BioPlex assays. Statistical significance evaluated using Wilcoxon matched-pairs signed-rank test, presented as mean with SD, significance levels—* *p* < 0.05, ns—non-significant.

**Figure 3 biomedicines-10-01476-f003:**
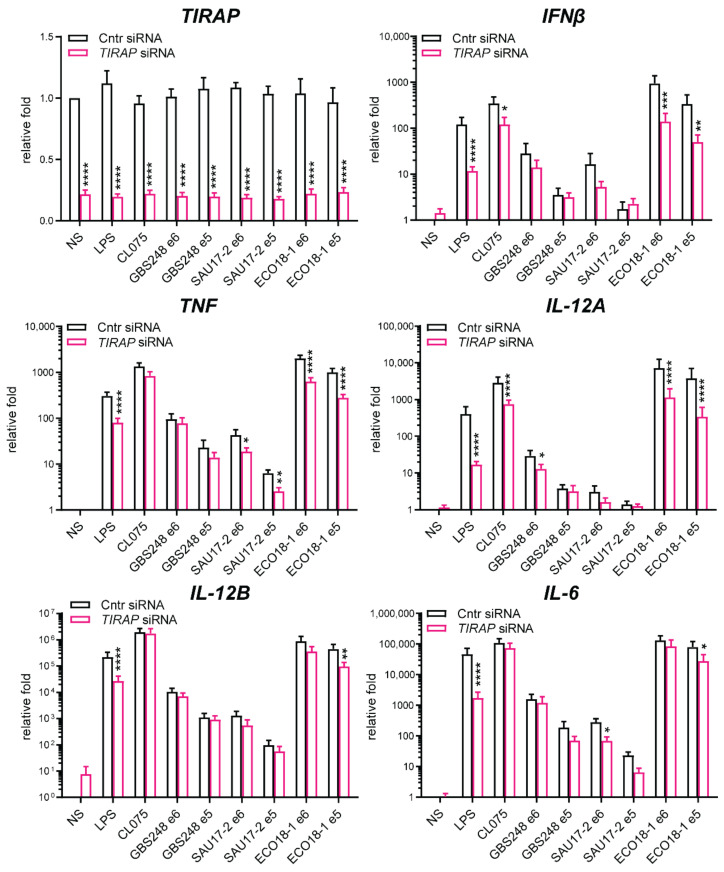
*TIRAP* silencing attenuates cytokine production from MDMs challenged with clinical isolates of *E. coli* (ECO), while affecting mainly *IL-12A* induction by Group B streptococcus (GBS) and pro-inflammatory cytokine induction by *S. aureus* (SAU). MDMs (5–6 donors in consecutive experiments) were pre-treated with *TIRAP* siRNA or control oligo and incubated with LPS (100 ng/mL), CL075 (1 µg/mL), or live bacteria (GBS 248, SAU 17-2, and ECO 18-1) for a total time of four hours. The doses of bacteria were 1 × 10^5^/mL (e5) and 1 × 10^6^/mL (e6). This roughly corresponds to MOI 0.01 and 0.1 for GBS, MOI 0.02 and 0.2 for SAU, and MOI 0.1 and 1.0 for ECO. Gene expression was determined by RT-qPCR, normalized to untreated sample, and presented as a mean relative fold change +SEM. Statistical testing was done with 2-way RM-ANOVA and post-test (* *p* < 0.05, ** *p* < 0.01, *** *p* < 0.001, **** *p* < 0.0001).

**Figure 4 biomedicines-10-01476-f004:**
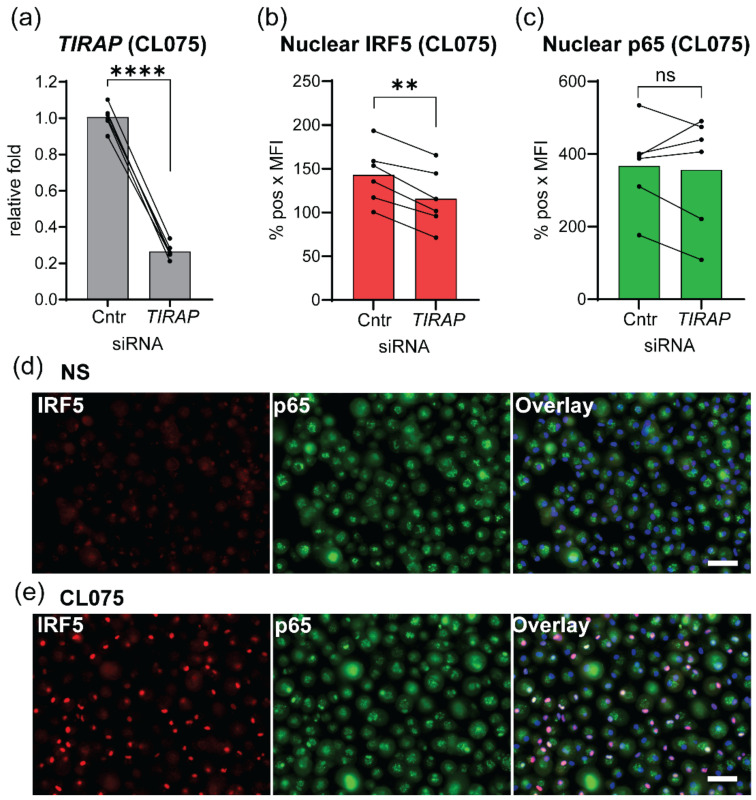
Silencing of *TIRAP* gene inhibits nuclear translocation of IRF5 in 60 min after CL075 stimulation. Experiments were performed on human MDMs (*n* = 6 donors). (**a**) *TIRAP*-silencing efficacy was quantified with RT-qPCR from parallel wells of CL075 stimulated cells using non-stimulated cells for normalization (fold = 1.0). Control or *TIRAP*-silenced cells were stimulated with CL075 (2 μg/mL) for one hour, followed by fixation of cells, double staining of IRF5 (**b**) and NF-kB (p65/RelA) (**c**), DNA staining by Hoechst 3342 for nuclei visualization, and quantitative imaging by high-content screening (Olympus Scan^R system). The level of nuclear IRF5 (**b**) and p65 (**c**) was calculated as the percentage of positively stained nuclei multiplied by the mean fluorescence-intensity value (MFI) of the positively stained nuclei. In non-stimulated cells, the background-staining levels (%pos × MFI) for nuclear IRF5 and p65 were <15 and <73, respectively. (**d**,**e**) Representative immunofluorescent images of non-stimulated (NS) and CL075 stimulated cells used for quantification of IRF5 (red channel) and p65 (green channel) in nuclei (blue channel). Scale bar shown in overlay represents 50 µm. Statistical significance was examined with paired *t*-test (** *p* < 0.01, **** *p* < 0.001).

**Figure 5 biomedicines-10-01476-f005:**
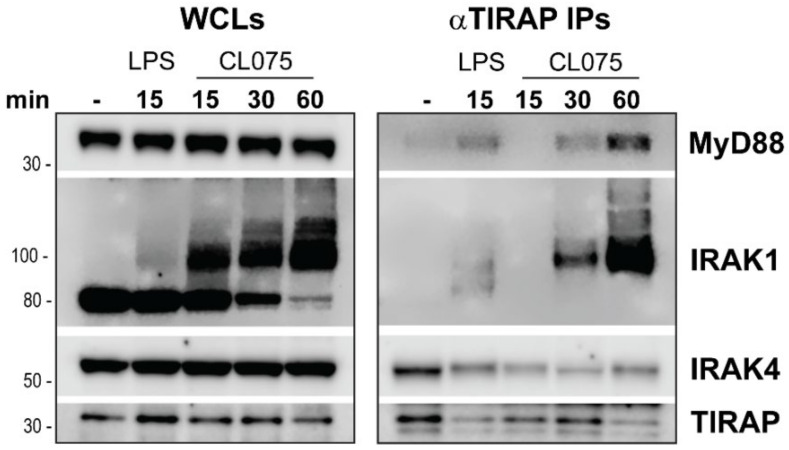
TIRAP is recruited to TLR8-initiated MyD88 and IRAK1/4-signaling complex. Endogenous TIRAP was immunoprecipitated for four hours from lysates (whole cell lysates—WCLs) of human MDMs: untreated or stimulated by LPS (100 ng/mL) or CL075 (2 µg/mL) for indicated time. LPS stimulation was applied as a positive control for TIRAP recruitment to the activated Myddosome. Cellular lysates were analyzed in parallel to control for input, with WB for MyD88, IRAK1, IRAK4, and TIRAP. A representative experiment is shown from a total of four consecutive experiments with different donors.

**Figure 6 biomedicines-10-01476-f006:**
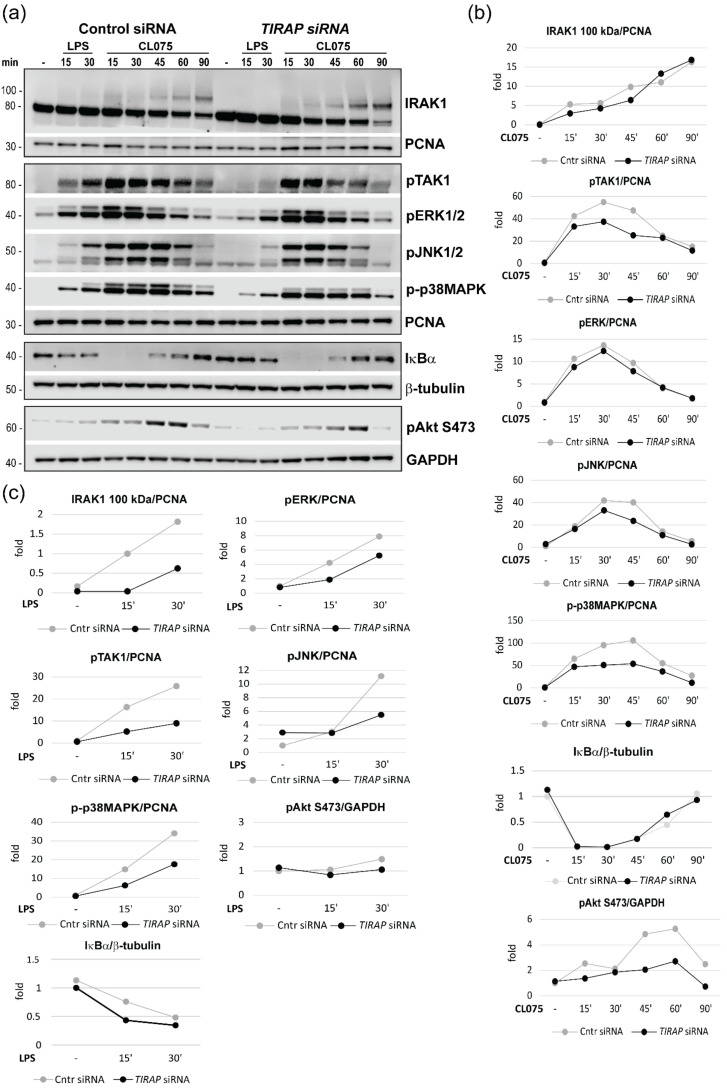
Silencing of *TIRAP* consistently inhibits TLR8-mediated phosphorylation of Akt S473. (**a**) Western blotting of lysates from MDMs treated with a control oligo or *TIRAP*-specific siRNA oligo and stimulated with 100 ng/mL LPS or 2 μg/mL CL075. The antibodies used are indicated on the figure, and GAPDH or PCNA are equal-loading controls. (**b**) Graphs show quantifications of protein levels relative to GAPDH or PCNA for CL075-stimulated cells and (**c**) LPS-stimulated cells. Representative image and graphs for one of four donors. Densitometry analysis and normalization to loading control was done using LiCor Odyssey software.

**Figure 7 biomedicines-10-01476-f007:**
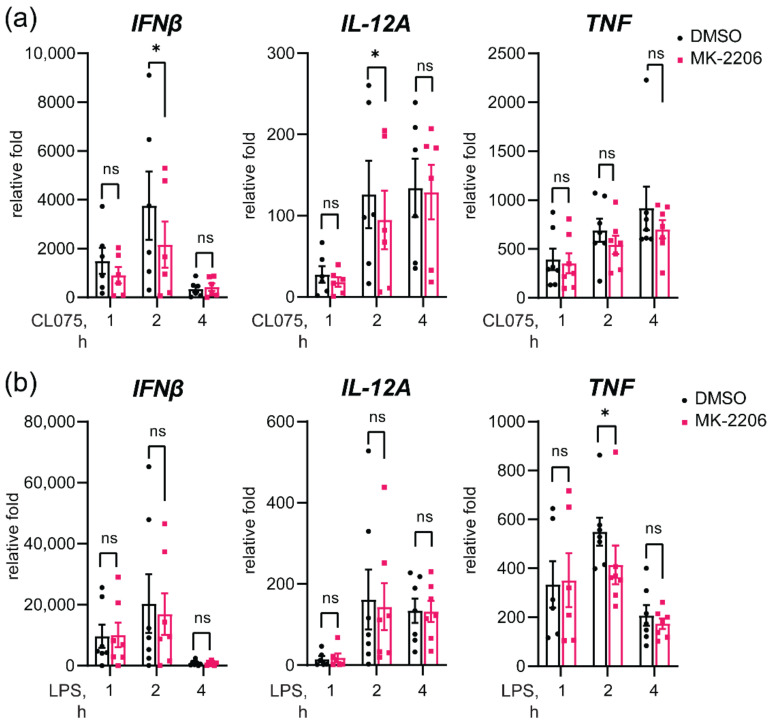
Akt inhibition in MDMs significantly reduces TLR8-mediated expression of *IFNβ* and *IL-12A* genes, while having no effect on TLR4-mediated cytokine expression. RT-qPCR analysis of cytokine expression after pre-treatment with Akt inhibitor MK-2206 (2 μM) and stimulation by (**a**) CL075 (2 µg/mL) or (**b**) LPS (100 ng/mL). Gene expression normalized to unstimulated sample and presented as a mean relative fold change +SEM. Statistical testing was done by 2-way RM-ANOVA including a post-test, as described (* *p* < 0.05, and ns—non-significant).

**Figure 8 biomedicines-10-01476-f008:**
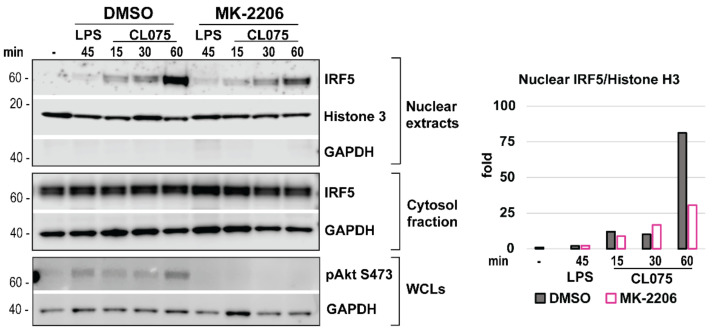
Inhibition of Akt reduced IRF5 nuclear translocation in human monocytes. Western blot analysis of cytosolic fraction and nuclear extracts from cells pre-treated with MK-2206 (2 µM) was followed by stimulation with CL075 (2 µg/mL) or LPS (100 ng/mL). LPS stimulation was applied for negative control. IRF5 levels in nuclear extracts were normalized based on Histone 3 bands’ intensity (graph), while GAPDH Western blot was performed to control for potential contamination of nuclear extracts with cytosol content. To control for Akt inhibition efficacy, Akt (S473) phosphorylation level was addressed in WCLs (whole cell lysates) from parallel wells. Representative of three consecutive experiments.

**Figure 9 biomedicines-10-01476-f009:**
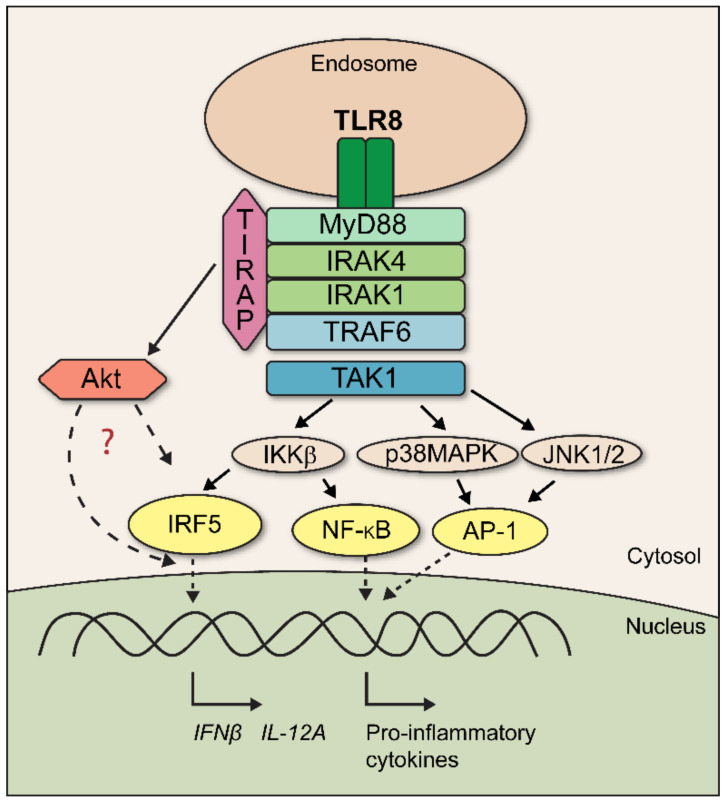
Model showing TIRAP involvement in regulation of TLR8 signaling. Recruitment of TIRAP to Myddosome promotes Akt activation and facilitates nuclear translocation of IRF5 as well as expression and secretion of IRF5-dependent cytokines IFNβ and IL-12p70.

## Data Availability

All the data for manuscript is provided in main text or Appendix A.

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
