# Peer review of "TIRAP/Mal Positively Regulates TLR8-Mediated Signaling via IRF5 in Human Cells"

_biomedicines, 2022, doi:10.3390/biomedicines10071476_

Round 1
Reviewer 1 Report
In the manuscript by Nilsen at al., the authors perform expansive experimentation to show how TLR8 signals an immune response following chemical stimulation or bacterial infection. They use siRNAs and chemical inhibition to block signaling events to test their hypothesis. The authors also use a wide variety of techniques as read-outs for their assays, providing rigor in their experimentation. Overall, the manuscript is well-written and the data are presented well. I have some minor suggestions that will nevertheless improve the clarity of data presentation, methods, and impact of the manuscript.
Line 186 and Figure 6 – how does the Odyssey software perform densitometry? Are bands normalized to a loading control? Are the data only representative of one experiment, and thus statistical tests are not performed on the data?
Figure 1 caption – TNF expression also appears to be down-regulated when TIRAP is knocked down, but this is not stated in the caption or associated text.
Lines 279-280 – TIRAP proteins levels appear to be decreased to me, so this statement seems incorrect. What is the fold change decrease of protein versus mRNA (Fig s1c) in these knockdown experiments?
Line 291 – define GBS in the main text.
Figure 3 – What does “e5” and “e6” represent on the x-axis? Also, for TIRAP expression, are there no significant differences among conditions?
Figure 4 – representative microscopy images should be shown for these data. A summary of mean +/- standard deviation for both IRF5 and p65 should be shown for clarity.
Figure 5 – since the author did no probe for interaction of TIRAP with TLR8 or TLR4, they show examine co-localization of the proteins. While TIRAP may interact with the MyD88osome during TLR8 stimulation, the authors do not definitively show that it interacts directly with TLR8, which is an important conclusion of this study. As reference, interaction between TIRAP and TLR4 was shown in the seminal study by Horng, Barton, and Medzhitov. (Nature Immunology 2001).
Lines 375-379 – Can the authors summarize the findings and state if these are the expected results based on previously published literature?
Reviewer 2 Report
The objective of this article is to analysis the involvement of TIRAP in TLR8-mediated signaling pathway. The message is that TIRAP is recruited in TLR8-MyDDosome where it regulates TLR8-mediated, Akt-dependent activation of IRF5 and subsequent expression of IFNβ and IL-12A. I think that the results presented do not support the conclusion and more experiments will be required.
· Figure 1: CL075 also induced a strong expression of the pro-inflammatory cytokine TNF, equivalently or even strongly than LPS and silencing of TIRAP significantly decreased both CL075 and LPS-induced expression of TNF in a similar manner. Could authors explain these results. The results in Figure 1 are expressed as “relative fold”. Could authors precise the legend? Are the “untreated sample” signal is normalized at 1 or 100 (or other)?
· Figure S3: could authors analyze the level of expression of unphosphorylated form of STAT1 in order to determine whether the observed decrease in pSTAT1 is related to a decrease in phosphorylation or protein level.
· Figure 3: TLR8 has been involved in signaling mediated by GBS and S. Aureus. However, only GBS248-induced IL-12A expression is affected by TIRAP siRNA. Silencing of TIRAP did not modulate signaling induced by GBS248 e5, SAU17-2 e6 and e5. Could authors explain these results. The authors should confirm their results by analyzing another endpoint such as STAT1 phosphorylation or IFNβ and IL-12A secretion.
· Figure 4: the authors should confirm their results by analyzing Iκ-Bα phosphorylation / degradation.
· IRF5 translocation is known to be associated the its phosphorylation. TIRAP siRNA (Figure 4) and Akt inhibition (Figure 8) decreased IRF5 translocation without affecting IRF5 phosphorylation. Could authors confirm the phosphorylation of IRF5 by using phospho-IRF5-specific antibody. While the authors investigated the phosphorylation of IRF5 in CL075-stimulated cells (Figure 4), they analyzed the nuclear translocation in untreated cells. Should authors analyze nuclear translocation in CL075-treated cells?
· Figure 5: LPS stimulation is used as a positive control for the assembly of myddosome. The kinetics for the expression and secretion of cytokines is similar for LPS and CL075 stimulation. Why the authors used different time of treatment of LPS and CL075 for the analysis of Myddosome assembly. The binding of MyD88 and IRAK1 to TIRAP is very weak in LPS treated sample compared to CL075. IRAK4 was found associated with TIRAP in basal condition and TIRAP-IRAK4 interaction seems decrease in LPS or CL075-stimulated cells. Would it be possible to blot TLR8 in order to ensure that the authors analyzed the TLR8-Myddosome. This is necessary to support their claim “TIRAP is attracted to the TLR8 Myddosome” (line 364). The authors indicated that the TLR8-Myddosome assembly is delayed compared to TLR4-Myddosome. However, the incubation time for LPS and CL075 is different. We can not rule out that the TLR4-Myddosome is more pronounced at 30-60 min with a kinetics similar to TLR8-Myddosome. TLR8-myddosome contained only modified form of IRAK1 in CL075-stimulated cells. Could authors discuss this point. The authors should add an IP control (beads) in order to ensure that IRAK4 did not bind the beads.
· Figure 5 showed a strong modification (ubiquitination / phosphorylation) of IRAK1 after 15 min of stimulation with CL075. The unmodified form of IRAK1 disappeared almost completely after 60min. In Figure 6, the modification is weaker and appeared after 45 min. This result could be related to the primary cells used in this assay. An internal control such as the 60 min (Figure 5) or 90 min (Figure 6) of stimulation with LPS would be useful to compared the ability of cells to activate IFN signaling.
· The authors claimed that the decreased TLR8-mediated phosphorylation of Akt in TIRAP-silenced cells is the most consistent effect. I can see a decrease in p38 phosphorylation that seems to me more important (similar to the decrease of phosphor-p38 in LPS-stimulated cells). It would be interesting to compare the effect of Akt inhibitor and p38 inhibitors on CL075-induced IFNβ expression. In figure 6, did the authors quantify multiple experiments? If yes, would it be possible to indicate the number of experiments and the standard deviation.
· The authors claimed that TIRAP “is recruited after formation of the proximal TLR8-Myddosome complex and enhances Akt/PKB activation” (line 442). Which experiments are supporting this statement? Complementary experiment will be required to support this claim.
· Figure 8. The authors should also analyze p38 phosphorylation. LPS did not activated IRF5 (nuclear translocation of IRF5) whereas it induced the expression and production of IFN and IL-12 and STAT1 phosphorilation. Could authors explain these results.
· In the discussion, the authors recognized that TIRAP silencing did not significant impact IFNβ and suggest that the power of statistical analysis is insufficient. It is not possible to conclude in that condition. The authors need to increase the number of samples in order to be able to perform a relevant statistical analysis.
